# Effect of Multiple Vaccinations with Tumor Cell-Based Vaccine with Codon-Modified GM-CSF on Tumor Growth in a Mouse Model

**DOI:** 10.3390/cancers11030368

**Published:** 2019-03-15

**Authors:** Jiantai Timothy Qiu, Donia Alson, Ta-Hsien Lee, Ching-Chou Tsai, Ting-Wei Yu, Yu-Sing Chen, Ya-Fang Cheng, Chu-Chi Lin, Scott C. Schuyler

**Affiliations:** 1Graduate Institute of Biomedical Sciences, College of Medicine, Chang Gung University, Taoyuan 33302, Taiwan; doniaalson86@gmail.com (D.A.); faekyle@gmail.com (T.-H.L.); stellarfumi@gmail.com (Y.-S.C.); lilian1122@hotmail.com (Y.-F.C.); chuchiyinghui@gmail.com (C.-C.L.); 2Department of Obstetrics and Gynecology, Chang Gung Memorial Hospital, Taoyuan 33302, Taiwan; 3Department of Biomedical Sciences, Chang Gung University, No. 259, Wenhua 1st Road, Kwei-Shan District, Taoyuan 33302, Taiwan; timmylife17@gmail.com (C.-C.T.); flywithfish@hotmail.com (T.-W.Y.); 4Division of Head & Neck Surgery, Department of Otolaryngology, Chang Gung Memorial Hospital, 5 Fushing Street, Kwei-Shan, Taoyuan 33302, Taiwan

**Keywords:** HPV, cGM-CSF, cervical cancer, multiple vaccinations, IKDCs

## Abstract

Ectopic expression of codon-modified granulocyte-macrophage colony-stimulating factor (cGM-CSF) in TC-1 cells (TC-1/cGM-CSF), a model cell line for human papillomavirus (HPV)-infected cervical cancer cells, increased the expression level of GM-CSF and improved the efficacy of tumor cell-based vaccines in a cervical cancer mouse model. The number of vaccine doses required to induce a long-term immune response in a cervical cancer mouse model is poorly understood. Here, we investigated one, three, and five doses of the irradiated TC-1/cGM-CSF vaccine to determine which dose was effective in inducing a greater immune response and the suppression of tumors. Our findings showed that three doses of irradiated TC-1/cGM-CSF vaccine elicited slower tumor growth rates and enhanced survival rates compared with one dose or five doses of irradiated TC-1/cGM-CSF vaccine. Consistently, mice vaccinated with three doses of irradiated TC-1/cGM-CSF vaccine exhibited stronger interferon gamma (IFN-γ) production in HPV E7-specific CD8^+^ T cells and CD4^+^ T cells. A higher percentage of natural killer cells and interferon-producing killer dendritic cells (IKDCs) appeared in the splenocytes of the mice vaccinated with three doses of irradiated TC-1/cGM-CSF vaccine compared with those of the mice vaccinated with one dose or five doses of irradiated TC-1/cGM-CSF vaccine. Our findings demonstrate that single or multiple vaccinations, such as five doses, with irradiated TC-1/cGM-CSF vaccine suppressed the immune response, whereas three doses of irradiated TC-1/cGM-CSF vaccine elicited a greater immune response and subsequent tumor suppression.

## 1. Introduction

Cervical cancer is the second most common cause of cancer-related deaths in women worldwide [1,2]. High-risk types of human papillomavirus (HPV), such as types 16 and 18, are often correlated with a higher percentage of cervical cancer [3,4]. Almost 99% of tissue samples from cervical cancer show positive indications of HPV infection [3,4,5,6]. The growth of HPV-positive cancer cells depends on the expression of two important viral oncogenes, E6 and E7 [7,8]. The continued expression of E6 and E7 oncogenes enhances the proliferation of infected cells and inactivates the function of two important tumor suppressor genes, p53 and pRb [9,10,11]. Because E6/E7 are continuously expressed in HPV-positive cancer cells, they serve as tumor-specific antigens that allow for the circumvention of immune tolerance against self-antigens and therefore provide an attractive therapeutic strategy for targeting HPV infection and HPV-associated malignancies [12,13].

Granulocyte-macrophage colony-stimulating factor (GM-CSF) is an immunomodulatory cytokine that can increase the immunogenicity of tumors and stimulate both humoral and cell-mediated immune responses [14,15]. Our previous report showed that mice vaccinated with irradiated TC-1 cells secreting codon-modified GM-CSF (cGM-CSF) exhibit enhanced survival rates by suppressing tumor growth [16]. Reports also show that multiple vaccinations with irradiated autologous melanoma cells engineered to secrete human GM-CSF (hGM-CSF) are a promising treatment for melanoma patients [17,18,19].

Currently available prophylactic bivalent and quadrivalent vaccines produced from virus-like particles (VLPs) effectively prevent infection from HPV type 16 and 18 [20,21]. Other vaccines, such as HPV L1 and HPV E6 protein vaccines, induce antigen-specific T cell responses, which are detectable by the proliferation of both CD4^+^ and CD8^+^ T cells, as well as the generation of memory T cells [22,23]. However, studies also show that repeated vaccinations with irradiated autologous melanoma cells engineered to secrete hGM-CSF or leukemia-associated antigens, such as PR1 and WT1 peptides, show no objective clinical response [24].

During HPV infection, multiple doses of HPV vaccine can provide protection for several years [25,26,27]. Three doses of bivalent vaccine provide protection against HPV infection by generating serum-neutralizing antibodies and a CD4^+^ T cell response; however, the same level of protection is also provided by a single dose of vaccine [28,29]. The question of whether one dose or multiple doses of TC-1/cGM-CSF vaccine can induce an effective immune response remains poorly understood in HPV-associated cervical cancer mouse models.

Because the risk of HPV infection is sustained throughout the life of an individual, it is important to know the period of protection HPV vaccinations provide. It has been established that tumor cell-based vaccines, such as TC-1/cGM-CSF, induce an immune response [16]; however, the effect of multiple vaccinations with TC-1/cGM-CSF on long-term immunity also remains poorly understood.

Thus, in this study, we evaluated the immune response of mice treated with irradiated TC-1 cells engineered to secrete cGM-CSF after one dose, three doses, and five doses of vaccine. We assessed the efficacy of different vaccination schedules for tumor suppression and overall mouse survival. Furthermore, we evaluated the generation of interferon gamma (IFN-γ)-secreting antigen-specific CD8^+^ T cells, CD4^+^ T cells, natural killer (NK) cells, and interferon gamma-producing killer dendritic cells (IKDCs) in mice vaccinated with one or multiple doses.

## 2. Materials and Methods

### 2.1. Mice

First, 6–8-week-old female C57BL/6 mice were obtained from the National Laboratory Animal Center (Taipei, Taiwan). All the mice were housed under specific pathogen-free conditions in the animal care facility of Chang Gung University. All the animal experiments were performed in accordance with the Animal Experimental Ethics Committee of Chang Gung University. The statement of approval from the ethical committee contains the ethical code “CGU11-135” with the date 16 October 2012 and the code “CGU15-010” with the date 1 April 2015.

### 2.2. Cell Lines

The TC-1 cells (ATCC No: CRL 2785TM) were engineered by the transformation of primary C57BL/6 lung cells with HPV type-16, E6 and E7 oncogenes, and an activated H-ras [30]. Stable TC-1 cells expressing wild-type granulocyte-macrophage colony-stimulating factor (wt-GM-CSF) or cGM-CSF were established by lentivirus infection of TC-1 cell lines (LV-TC-1-cGM-CSF) [31,32,33]. All the cell lines (i.e., TC-1, TC-1/wt-GM-CSF, and TC-1/cGM-CSF) were maintained in RPMI-1640 medium (Gibco, Waltham, MA, USA) supplemented with 2 mM L-glutamine, 25 mM HEPES, 24 mM sodium bicarbonate, 10% heat-inactivated fetal bovine serum (Invitrogen, Waltham, MA, USA), 100 U/mL penicillin, 100 mg/mL streptomycin, and 50 µM β-mercaptoethanol at 37 °C with 5% CO_2_.

### 2.3. Cytokine Secretion Measurement with ELISA

GM-CSF levels from the secreted cells were measured by ELISA with a commercially available GM-CSF ELISA kit (R&D Systems, Minneapolis, MN, USA). A total of 1 × 10^6^ cells (TC-1, TC-1/wt-GM-CSF, and TC-1/cGM-CSF) were cultured in 7 mL of medium for 24 h in a 10 cm dish. Supernatants were collected by centrifugation at 220 RCF (Relative centrifugal force) for 4 min after 24 h and diluted appropriately.

### 2.4. Tumor Model and Vaccination

For the in vivo tumor protection experiment, C57BL/6 mice (*n* = 10 per group) were immunized subcutaneously in the dorsal flank with 1 × 10^6^ irradiated (10,000 cGy) TC-1, TC-1/wtGM-CSF, and TC-1/cGM-CSF at days 0, 14, 28, 42, and 56. Then, 7 days after the last vaccination, the immunized mice were subcutaneously challenged with 5 × 10^5^ TC-1 cells in the right dorsal flank. Tumor growth was monitored 3 times a week using calipers and was calculated according to the formula: length × (width)^2^ × 0.5. When the tumor growth exceeded 2 cm in diameter, the mice were considered dead from the tumor burden and were subsequently euthanized. For the immune cell analysis, mice were subcutaneously immunized in the dorsal flank with 1 × 10^6^ irradiated (10,000 cGy) TC-1, TC-1/wtGM-CSF, and TC-1/cGM-CSF at days 0, 14, 28, 42, and 56. Then, 7 days after the last vaccination, spleens were collected from the mice for flow cytometric analysis (Appendix A).

### 2.5. Flow Cytometric Analysis

To analyze intracellular IFN-γ production by CD8^+^ and CD4^+^ T cells, splenocytes from the mice vaccinated with 1, 3, and 5 doses of cGM-CSF were collected 7 days after the last immunization and stimulated in vitro with 10 µg/mL HPV-16 E7 MHC class I peptide (aa 49–57, RAHYNIVTF) or MHC II (aa 44–60, QAEPDRAHYNIVTFCCK) peptide by incubation at 37 °C with 5% CO_2_ for 15 h. After 15 h, the cells were treated with 50 ng/mL phorbol-12-myristate-13-acetate (PMA) and 1 µg/mL of ionomycin (Sigma-Aldrich, St. Louis, MO, USA) in the presence of GolgiStop protein transport inhibitor containing monensin (BD Bioscience, San Jose, CA, USA) for 4 h. The cultured cells were analyzed by flow cytometry.

For IKDC analysis, splenic cells were stained with anti-B220-FITC (BD Bioscience, San Diego, CA), anti-NK 1.1-perCP (eBioscience, San Diego, CA, USA), anti-TCRβ-PE (eBioscience, San Diego, CA, USA), anti-CD19-PEcy7 (eBioscience, San Diego, CA, USA), and anti-CD11c-APC antibodies (Biolegend, San Diego, CA, USA) at 4 °C for 20 min followed by flow cytometric analysis.

### 2.6. Statistical Analysis

All the analyses were performed using GraphPad Prism statistical software (Graph Pad Software, La Jolla, CA, USA). Two-way ANOVA and log-rank (Mantel-Cox) tests were used to analyze the tumor growth and mouse survival data, respectively. All the other data were analyzed using unpaired two-tailed *t*-tests. A value of *p* < 0.05 was considered statistically significant.

## 3. Results

### 3.1. TC-1 Cells Transfected with LV-cGM-CSF(Lentiviral) (TC-1/cGM-CSF) Expressed Increased Levels of GM-CSF Compared with TC-1 Cells Transfected with LV-wtGM-CSF (TC-1/wtGM-CSF)

Our previous study showed that vaccination with irradiated TC-1 cells overexpressing cGM-CSF resulted in stronger IFN-γ production, more dendritic cells recruitment to draining lymph nodes (dLNs), and enhanced overall survival of the mice. Therefore, this approach improves the efficacy of tumor cell-based vaccines for cancer immunotherapy [16]. To understand whether multiple vaccinations with cGM-CSF can augment an effective immune response, we generated lentiviral vectors that expressed wt-GM-CSF and cGM-CSF to infect TC-1 cells. The transfected cells were grown in culture media, and medium containing GM-CSF was collected from the cells cultured for 24 h to perform ELISA. As shown in Figure 1, TC-1 cells infected with LV-cGM-CSF produced increased levels of GM-CSF compared with the TC-1 cells infected with LV-wtGM-CSF. These results show that GM-CSF is expressed more effectively when its codons are modified.

### 3.2. Mice Vaccinated with Three Doses of Irradiated TC-1/cGM-CSF Induced Enhanced Immunosurveillance Compared with Mice Vaccinated for TC-1 Tumor with One Dose and Five Doses

Our previous study showed that vaccinating with irradiated TC-1 cell overexpressing cGM-CSF resulted in a significant reduction in tumor growth and extended overall survival [16]. To check the effect of multiple vaccinations with one, three, and five doses of the vaccine on tumor suppression and overall survival, mice were vaccinated via subcutaneous injection, separated by a two-week interval with irradiated TC-1/cGM-CSF. Tumor volume was measured twice weekly. We observed inhibition of subcutaneous tumor growth in C57BL/6 mice challenged with TC-1 cells one week after the three doses of vaccination with irradiated TC-1/cGM-CSF. In contrast, animals vaccinated with either one dose or five doses of TC-1/cGM-CSF vaccination revealed rapid subcutaneous tumor growth (Figure 2A). We found that vaccination with three doses of irradiated TC-1/cGM-CSF showed TC-1 tumor regression in 90% of the mice, and these mice remained tumor free over the test period. In contrast, mice vaccinated with one dose and five doses of irradiated TC-1/cGM-CSF vaccine showed tumor regression in 0% and 60% of the mice, respectively (Figure 2A). We also found an increase in the life span of C57BL/6 mice (*n* = 23) vaccinated with three doses of irradiated TC-1/cGM-CSF vaccine and challenged after one week with subcutaneous injection of TC-1 cells compared with those of the groups of mice vaccinated either one time (*n* = 20) (*p* < 0.001) or five times (*n* = 23) (*p* < 0.01) with irradiated TC-1/cGM-CSF vaccine. These results showed that mice vaccinated with three doses of irradiated TC-1/cGM-CSF vaccine displayed the most efficient TC-1 tumor regression compared with mice that had been vaccinated with one or five doses of irradiated TC-1/cGM-CSF vaccine.

### 3.3. Generation of Antigen-Specific IFN-γ-Producing CD8^+^ and CD4^+^ T Cells

CD8^+^ T cells have a recognized role in anti-tumor immunity; hence, we evaluated the CD8^+^ T cell immune response. Splenocytes from the mice vaccinated with one dose, three doses, and five doses of irradiated TC-1/cGM-CSF vaccine were collected and stimulated in vitro with HPV-16 E7 MHC class I peptide (aa 49–57, RAHYNIVTF), and intracellular IFN-γ production was analyzed by flow cytometry. Five doses of vaccination generated fewer E7-specific IFN-γ-containing CD8^+^ T cells compared with one dose and three doses of irradiated TC-1/cGM-CSF vaccination (Figure 3A). These data indicate that five doses of vaccination stimulate a lesser E7-specific T cell response compared with one and three doses of irradiated TC-1/cGM-CSF vaccination.

We also evaluated the CD4^+^-specific T cell immune response generated after one dose, three doses, and five doses of irradiated TC-1/cGM-CSF vaccine. Splenocytes from vaccinated mice were stimulated in vitro with HPV-16 E7 MHC class II peptide (aa 44–60, QAEPDRAHYNIVTFCCK), and intracellular IFN-γ levels were analyzed by flow cytometry. As shown in Figure 3B, IFN-γ levels were enhanced in the group of mice receiving one dose and three doses of cGM-CSF vaccination compared with those of mice receiving five doses. These data indicate that one dose and three doses of vaccination stimulate the E7-specific T cell response in mice compared with five doses of irradiated TC-1/cGM-CSF vaccination (Figure 3B). Spleens were collected and weighed from different groups of vaccinated mice, and the results show that the group of mice vaccinated with three doses and five doses of irradiated TC-1/cGM-CSF vaccine weighed less than the mice vaccinated with one dose of irradiated TC-1/cGM-CSF vaccine, indicating dramatically suppressed immune cell proliferation (Figure 3C). Splenic CD4^+^ T cells expressing the transcription factor Foxp3 were identified using intracellular immunofluorescence staining seven days after the last vaccination. We observed no statistically significant difference in the percentage of CD4^+^ CD25^+^ Foxp3^+^ cells in mice vaccinated either one time (5.76%), three times (3.83%), or five times (4.75%) with irradiated TC-1/cGM-CSF vaccine (Figure 3D).

### 3.4. A Higher Percentage of B220^+^ NK1.1^+^ IKDC Were Produced in Mice Vaccinated with Three Doses of Irradiated TC-1/cGM-CSF Vaccine Compared with Those of Mice Vaccinated with One Dose and Five Doses of Irradiated TC-1/cGM-CSF Vaccine

Natural killer cells are known to act against cancer cells and perform a lytic function through granzyme B and perforin release [34]. Previous studies have shown that a new subset of cells called IKDCs, which exhibit the properties of both natural killer cells and dendritic cells, play an important role in tumor immunosurveillance [35]. Mice were vaccinated and boosted with one, three, or five doses of irradiated TC-1/cGM-CSF vaccine fortnightly. Seven days after the last vaccination, cells from the spleen were collected and analyzed by flow cytometry for IKDCs. IKDCs were gated for the population negative for T cell receptor β, B cell surface marker CD19, and the population positive for natural killer cells (NK1.1^+^, B220^+^) and dendritic cell marker, CD11c^dim^ (TCRβ^−^ CD19^−^ NK1.1^+^ CD11c^dim^ B220^+^). The results show that the percentages of IKDCs from the splenic cells appeared to be higher in the group of mice vaccinated with three doses (52.3%) of irradiated TC-1/cGM-CSF vaccine compared with those of the groups of mice vaccinated with one (34.2%) or five doses (13.2%) of TC-1/cGM-CSF vaccine (Figure 4). The data also show a significant increase in the percentage of natural killer cells in the group of mice vaccinated with three doses (14.8%) of irradiated TC-1/cGM-CSF vaccine compared with those of mice vaccinated with one dose (1.49%) or five doses (3.33%) of irradiated TC-1/cGM-CSF vaccine (Figure 4). These results showed that five doses of vaccine with cGM-CSF reduced the generation of IKDCs, and hence, an optimal vaccination dose is required for higher IKDC generation. There were no adverse effects on the host when irradiated TC-1/cGM-CSF vaccines were injected subcutaneously multiple times.

## 4. Discussion

This study showed that in a cervical cancer mice model [30] vaccination with three doses of irradiated TC-1/cGM-CSF vaccine regressed tumor growth and enhanced the overall survival of mice compared with that of mice receiving five doses of irradiated TC-1/cGM-CSF vaccine. We also found that three doses of vaccination induced higher IFN-γ-secreting CD8^+^ and CD4^+^ T cell responses compared with those in mice receiving five doses of vaccination. Lastly, we showed that vaccination with three doses of irradiated TC-1/cGM-CSF vaccine generated a higher percentage of IKDCs compared with that in mice receiving five doses of irradiated TC-1/cGM-CSF vaccine. We previously showed that mice vaccinated with three doses of irradiated TC-1/cGM-CSF tumor cell-based vaccine exhibited stronger IFN-γ production, higher dendritic cell recruitment in draining lymph nodes, and an enhanced survival rate in a mouse model [16]. Interestingly, our vaccine schedule in this study showed a greater immune response with three doses of irradiated TC-1/cGM-CSF vaccination compared with that of five doses of vaccination.

Multiple doses of HPV vaccines are effective in providing protection against HPV-associated cervical cancer by inducing CD4^+^ and CD8^+^ T cell immune responses for several years [22,23,28,29]. Additionally, a single dose of bivalent HPV vaccine provides a similar level of protection as a three- dose vaccine schedule [36,37]. The question of how many doses of vaccination are best for HPV immunization needs to be answered. Currently available bivalent and quadrivalent vaccines provide protection against cervical cancer and genital warts for four different types of HPVs (Types-6, -11, -16, and -18), but the risk of cervical cancer still occurs because of the possibility of infection with other HPV types [21,38].

It is well known that cytokines help stimulate immune effector cells and enhance tumor cell recognition by cytotoxic effector cells [39,40]. Proinflammatory cytokines, such as IFN-γ, are known to be involved in cytotoxic and anti-tumor mechanisms during cell-mediated adaptive immune response [41,42,43]. In this study, vaccination with three doses of irradiated TC-1/cGM-CSF vaccine evoked a higher E7-specific cellular immune response compared with five doses of vaccination, which is likely to have contributed to the anti-HPV tumor effects observed after prophylactic vaccination. We observed in vitro stimulation of E7-peptide or induced IFN-γ production by CD8^+^ or CD4^+^ T cells, respectively, in immunized mice, which implies the role of CD8^+^ T cells and CD4^+^ T cells in anti-tumor activity. Although studies show the role of T regulatory (Treg) cells in suppressing the efficacy of the antitumor GM-CSF vaccine against B16 melanoma [44], the changes in the Treg frequencies in the present study were not significant after one dose, three doses, and five doses of irradiated TC-1/cGM-CSF vaccination. Studies shows that myeloid-derived suppressor cells (MDSCs) and tolerogenic monocytes are also involved in tumor immune suppression in addition to T regulatory cells [45]. Thus, further studies are needed to understand the role of myeloid-derived suppressor cells and tolerogenic monocytes in immune suppression after multiple vaccination with irradiated TC-1/cGM-CSF vaccine.

Recently, new hybrid immune cell types were identified as IKDCs, which resemble both NK cells and DCs, and were originally described as CD11c^int^ B220^+^ NK1.1^+^ CD3^−^ CD19^−^ Gr-1^−^ cells [45,46]. IKDCs exert potent anti-tumor activity through the secretion of IFN-γ and the direct killing of tumor cells [47,48,49]. Data from this study show a significant increase in the percentage of IKDCs that might be involved in anti-tumor activity in the group of mice vaccinated with one dose or three doses of irradiated TC-1/cGM-CSF vaccine compared with that in the group of mice vaccinated with five doses of irradiated TC-1/cGM-CSF vaccine. This implies that five doses of vaccination downregulate the generation of IKDCs compared with three doses. A higher percentage of IKDC generation followed by tumor suppression shows that IKDCs might be playing an important role in anti-tumor activity. Further studies are needed to establish the role of IKDCs in TC-1/cGM-CSF vaccine-induced anti-tumor activity.

In summary, we demonstrated that vaccination with three doses of irradiated TC-1/cGM-CSF vaccine can elicit a greater E7-specific anti-tumor immune response in mice compared with those in mice receiving one dose or five doses of irradiated TC-1/cGM-CSF vaccine. Additionally, there was a significant increase in the level of IKDCs after three doses of vaccination compared with that after one dose and five doses of vaccination. 

## 5. Conclusions

In conclusion, in the group of mice vaccinated one time, three times, and five times, we have shown that three doses of irradiated TC-1/cGM-CSF vaccination elicited a stronger immune response, slower tumor growth rates, and enhanced survival rates compared with one dose or five doses of irradiated TC-1/cGM-CSF vaccination. Mice vaccinated with three doses of irradiated TC-1/cGM-CSF vaccination exhibited stronger IFN-γ production in HPV E7-specific CD8^+^ T cells and CD4^+^ T cells. A higher percentage of IKDCs appeared in the splenocytes of the mice vaccinated with three doses of irradiated TC-1/cGM-CSF compared with those of the mice vaccinated with one dose or five doses of irradiated TC-1/cGM-CSF vaccine. This result suggests that an optimum number of irradiated TC-1/cGM-CSF vaccinations are required for enhanced immune response and subsequent tumor suppression. Further studies are needed to produce a better understanding of the total number of irradiated TC-1/cGM-CSF vaccinations required to fight HPV-induced cervical cancer in other animal models, such as in primates, and also in human beings.

## Figures and Tables

**Figure 1 cancers-11-00368-f001:**
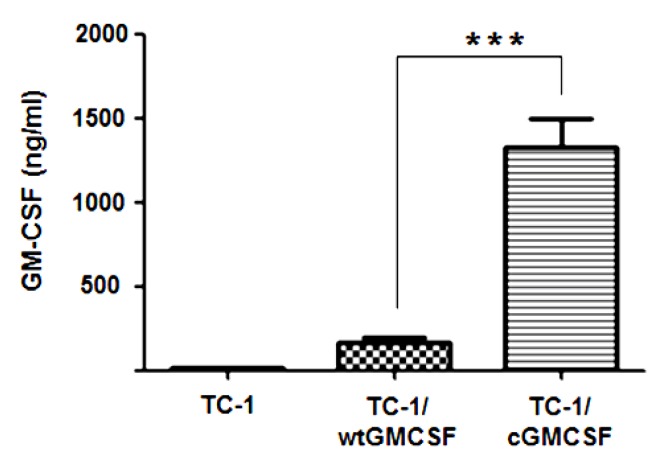
Increased levels of granulocyte-macrophage colony-stimulating factor (GM-CSF) production by TC-1 cells containing codon-modified GM-CSF. The level of GM-CSF production was quantified by performing ELISA. The results shown are representative of three independent experiments. *** *p* < 0.001; single classification ANOVA. wtGM-CSF: wild-type GM-CSF; cGM-CSF: codon-modified GM-CSF.

**Figure 2 cancers-11-00368-f002:**
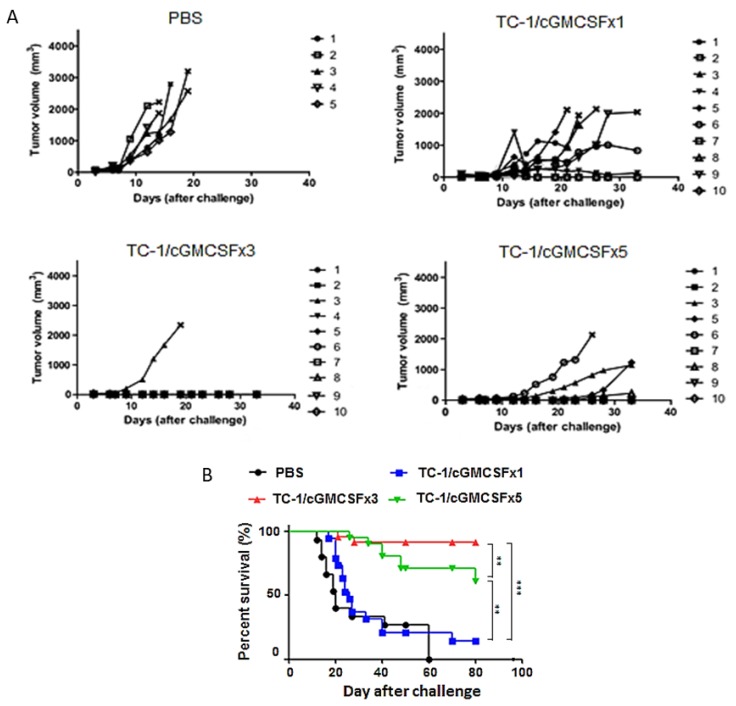
Tumor vaccination three times with cGM-CSF can more efficiently inhibit tumor growth compared with five-time vaccination. (**A**) C57BL/6 mice were injected with 1 × 10^6^ irradiated TC-1/cGM-CSF subcutaneously at in the dorsal flank. Seven days after the last immunization, with a time interval of two weeks for multiple vaccinations, the mice were inoculated with 5 × 10^5^ TC-1 cells subcutaneously at the right flank. The tumor size was monitored twice weekly for 6 weeks. The line graphs depict tumor volumes over time in various vaccinated mouse groups. The growth curve is shown in panel A. (**B**) Survival curve with mice vaccinated with PBS (*n* = 15), one dose of irradiated TC-1/cGM-CSF vaccine (*n* = 20), three doses of irradiated TC-1/cGM-CSF vaccine (*n* = 23), and five doses of irradiated TC-1/cGM-CSF (*n* = 23) vaccine. The log-rank (Mantel-Cox) test was used to compare the survival rates among various groups. ** *p* < 0.01, *** *p* < 0.001 (log-rank (Mantel-Cox) test).

**Figure 3 cancers-11-00368-f003:**
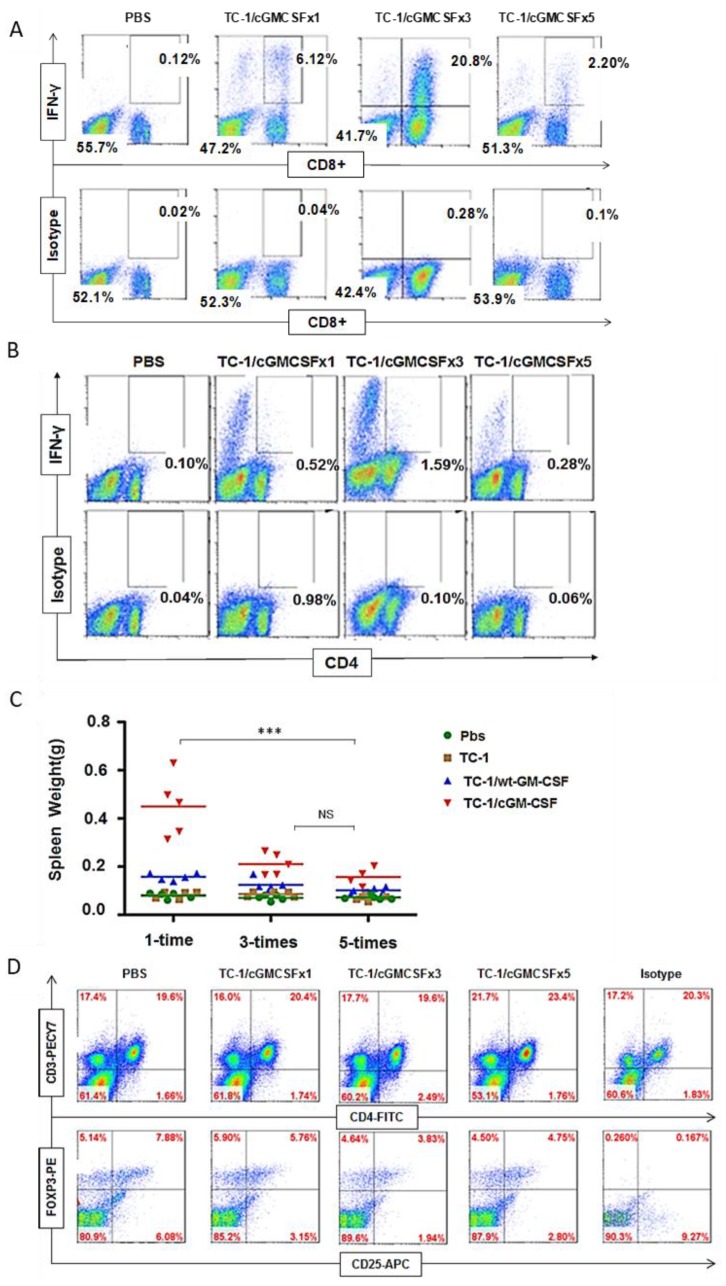
Vaccinating three times with cGM-CSF generates a higher IFN-γ-secreting CD8^+^ T and CD4^+^ T cell response compared with five-time vaccination. C57BL/6 mice were subcutaneously immunized with irradiated TC-1/cGM-CSF in the dorsal flank. Seven days after the last immunization, cells were isolated from spleens and re-stimulated with 10 µg/mL human papillomavirus (HPV)-16 E7 MHC class II peptide. IFN-γ-secreting cells (**A**) CD8^+^ T and (**B**) CD4^+^ T cells after one time, three times, and five times vaccinations were analyzed. (**C**) Spleen weight was analyzed after vaccination one time, three times, and five times. (**D**) FoxP3-positive (Regulatory T cell (T_reg_)) subsets were analyzed after vaccination one time, three times, and five times with irradiated TC-1/cGM-CSF. The data are presented as the mean SD of triplicate values. *** *p* < 0.001; single-classification A2NOVA.

**Figure 4 cancers-11-00368-f004:**
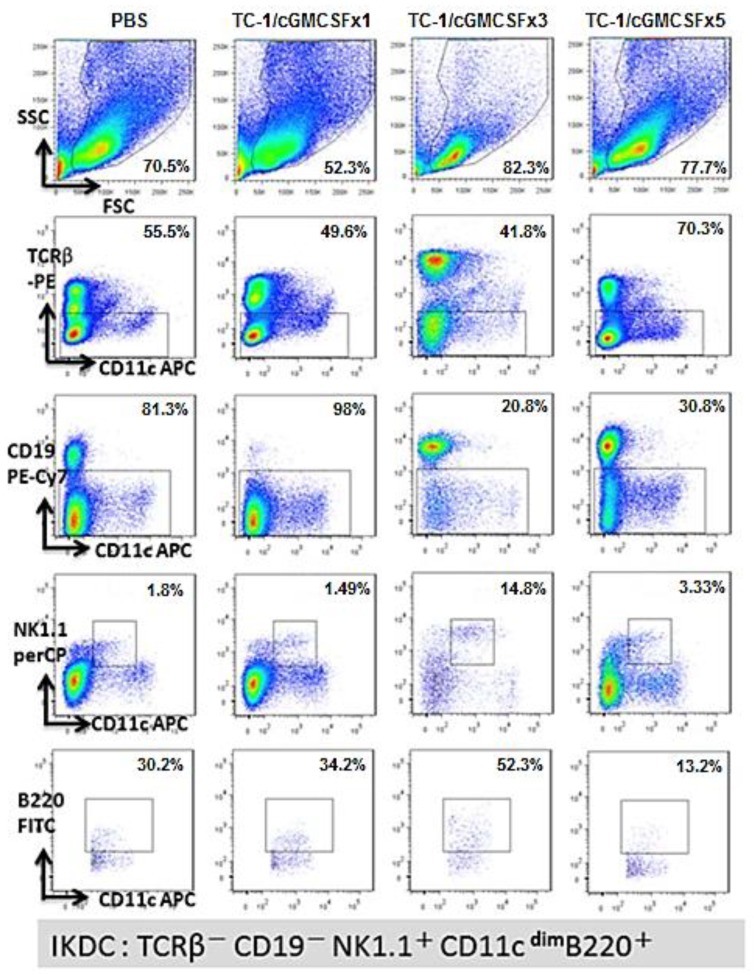
Vaccination three times with irradiated TC-1/cGM-CSF vaccine generates a higher percentage of interferon-producing killer dendritic cells (IKDCs) compared with vaccination five times. C57BL/6 mice were subcutaneously immunized with 1 × 10^6^ irradiated TC-1/cGM-CSF cells in the dorsal flank. Seven days after the last immunization, cells were isolated from the mice’s spleens, and flow cytometry was performed to characterize IKDCs markers CD11c^dim^, B220^+^, NK1.1^+^, TCRβ^−^, and CD19.

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
