# Peer review of "Effect of Multiple Vaccinations with Tumor Cell-Based Vaccine with Codon-Modified GM-CSF on Tumor Growth in a Mouse Model"

_cancers, 2019, doi:10.3390/cancers11030368_

Round 1
Reviewer 1 Report
In this manuscript Qiu et al. describe differences in protection against a sc TC-1 tumor challenge in B6 mice ,after 1, 3 or 5 administrations (at a 2-week interval) of a tumor vaccine. The vaccine consists of irradiated TC1 cells expressing either codon optimized or non-codon optimized GM-CSF. Firstly, they demonstrate that codon optimization results in improved GM-CSF production in TC-1 cells. Next, they show that protection after 3 administrations of the optimed GM-CSF expressing irradiated TC-1 cells is superior to that after 5 vaccinations or a single vaccination. This finding correlates with HPV specific T-cell counts in peripheral blood and with IKDC counts in peripheral blood. The finding that increased boosting results in decreased vaccine efficacy is interesting. It may be relevant in future anti-cancer vaccinations.
General comments:
Although IKDC may play a role, a causal relationship between IKDC levels and superiority of the 3x regimen is not established. This would require additional experiments, such as depletion of these cells.
The T (and B) lymphocyte pool in the peripheral blood after 3 administrations is at its maximum expansion (deferred from figure4) The percentage of TCRB pos cells falls from approximately 60 to 30% after 2 additional administrations (ie 5x regimen). Over the same timespan the E7 specific CD8 Tcells fall from 20 to 2%. This suggest that non-specific T cells have expanded as well due GM-CSF exposure. This and other pleiotropic effects (effects on Bcells and innate immune cells) of GM-CSF may all contribute to the superiority of 3x regimen.
if
it would help if it were clear to what extent the superiority of 3x
cGM-CSF/TC1 vaccination is due to E6/E7 specific immunity. Does the
phenomenon that an increased number of administrations results in
decreased vaccine efficacy hold up in a setting where wt-GM-CSF is
used rather than cGM-CSF? Can this phenomenon be
replicated by sc GM-CSF administrations combined with other means of
vaccination?
comments on the text
In the introduction and the discussion the information about various regimens of HPV VLP vaccination distract from the topic of the paper, as it relates to humoral immunity rather then cellular immunity, which is the subject of all findings presented in this paper.
The first phrase of the discussion mentions : " this study showed ... vaccination against HPV associated cervival cancer " . Clearly, the TC-1 line is not a cervical cancer...
The last phrase of the second-last paragraph of the discussion mentions : "A higher percentage of IKDC ..... shows the role of IKDC in anti-tumor immunity ..." The work presented does not justify such a statement.
The last phrase of the last paragraph of the discussion mentions: This implies that multiple booster vaccines do not ..... down regulate immune cells" For most vaccines immunity improves when the number of administration is increased. The statement should be rephrased keeping that in mind.
Author Response
Dear Reviewer,
We are writing in response to the comments on our manuscript ID: Cancers-438963, Title: Effect of multiple vaccinations with tumor cell-based vaccine with codon modified GM-CSF on tumor growth in a mouse model.
We appreciate the careful review and constructive suggestions. It is our belief that the manuscript is substantially improved after making the suggested edits.
We have attached here the word document with the response to the comments.

Reviewer 2 Report
Qiu et al., evaluated of TC-1/cGM-CSF vaccine dose required to induce long-term immune response in a cervical cancer mouse model. The study shows three but not one or five doses of TC-1/cGM-CSF slows tumor growth and enhances survival rate. Mice receive three doses of TC-1/cGM-CSF exhibit stronger IFN-γ production in HPV E7-specific CD8+ T cells and CD4+ T cells, and higher interferon-producing killer dendritic cells (IKDCs) splenocytes. The manuscript is well written and the data supports conclusion.
Comments:
Fig. 2A: Complete tumor regression was observed in several mice received one and five doses of vaccine. Any explanation?
Author Response

(The authors gave the same response as above.)

Reviewer 3 Report
Immunotherapy for cervical cancer is presently considered an effective preventive treatment method and there are multiple drugs in the market. But the present available strategies have limitations and there is always a need for new strategies. The present authors have shown in their previous studies that irradiated codon modified GM-CSF (cGM-CSF) in TC-1 cells can induce immune system against HPV mediated cervical carcinoma in m ice model. The present article is only an extension of their previous work where the authors have assessed the optimal number of dosages of vaccination to obtain the most effective dose. The authors investigated 1, 3 and 5 repeating dosages of vaccinations in 2 weeks apart and successfully demonstrated that 3 repeating dose is optimal for most effective therapeutic response. The optimal dose determination is an important step for any therapy and hence, the present work is important. But the article does not demonstrate sufficient novelty to match the standard of the journal. Moreover, though the experiments were performed thoroughly, the explanations and discussions of the results require more elaboration. Considering these points, with due respect to the hard works of the authors, I recommend the present article to be reconsidered after major revision. Following are the suggestions for the authors:
i. Though theoretically INF-γ is associated to both CD8+ and CD4+ T cells, According to Figure 3, its association to CD4 is negligible compared to CD8. Hence the comparison of CD4 response between groups with respect to INF-γ falls within error range in any flow-cytometric analysis. This is also evident with the low % numbers obtained in Figure 3B. Please use a different marker other than INF-γ to verify the CD4+ T cell production. Some anti-CD4 antibody can be considered for example.
ii. Though the authors have shown CD3/CD4 (Fig 3D), TCRβ/CD11 and CD19/CD11 (Fig 4) flow channel results, they have not explained or discussed in the main text. These results should be properly explained and discussed in the right context.
iii. The authors proved that an immune-suppression is going on at higher dose and the immune suppression in the higher dosages is not due to the Treg. It would increase the merit of the work if the authors can include a mechanistic insight of the immune-suppression by means of experiment or elaborated discussion with relevant references.
iv. Many abbreviations have been introduced (e.g. wtGM-CSF) without mentioning the full form of it. According to the journal policy, please explain the full name whenever an abbreviation is introduced for the 1st time.
Author Response

(The authors gave the same response as above.)

Round 2
Reviewer 3 Report
The authors have provided satisfactory answers to the reviewers’ questions and made necessary changes in the manuscript according to the suggestions. Hence I would suggest the manuscript to be published in its present form.